# Reactive-Sputtered Prepared Tin Oxide Thin Film as an Electron Transport Layer for Planar Perovskite Solar Cells

**Wenhai Sun [1], Shuo Wang [1], Shina Li [1], Xu Miao [1], Yu Zhu [1], Chen Du [1], Ruixin Ma [1,2,*] and Chengyan Wang [1,3,*]**

[1] School of Metallurgical and Ecological Engineering, University of Science and Technology Beijing, Beijing 100083, China; g20178246@xs.ustb.edu.cn (W.S.); b20180139@xs.ustb.edu.cn (S.W.); s20180252@xs.ustb.edu.cn (S.L.); g20178237@xs.ustb.edu.cn (X.M.); g20178277@xs.ustb.edu.cn (Y.Z.); s20170220@xs.ustb.edu.cn (C.D.)

[2] Beijing Key Laboratory of Special Melting and Preparation of High-End Metal Materials, University of Science and Technology Beijing, Beijing 100083, China

[3] Beijing Key Laboratory of Rare and Precious Metals Green Recycling and Extraction, University of Science and Technology Beijing, Beijing 100083, China

* Correspondence: maruixin@ustb.edu.cn (R.M.); q1590q@163.com (C.W.); Tel.: +86-010-623-33170 (R.M.)

**Abstract:** Currently, tin oxide ($SnO_2$) is a highly sought-after semiconductor material used in perovskite solar cells (PSCs) because of its good transmittance, the appropriate energy level, high electron mobility, high conductivity, ideal band gap and excellent chemical stability. In this study, $SnO_2$ film was successfully prepared by radio frequency reactive magnetron sputtering (RS) under room temperature conditions. The obtained $SnO_2$ thin films not only exhibited high transmittance in the visible region as well as the pure phase, but also had a suitable energy band structure and lower surface roughness than FTO ($SnO_2$:F) glass substrate, which contributes to the improvement of the adjacent interface morphology. The $SnO_2$ films prepared by reactive sputtering could effectively suppress carrier recombination and act as an electron transport layer. Moreover, the maximum efficiency of the device based on reactive sputtering of $SnO_2$ as the electron transport layer (ETL) for planar perovskite solar cells (PSCs) was 14.63%. This study mainly described the preparation of $SnO_2$ by reactive sputtering under room temperature conditions.

**Keywords:** $SnO_2$ thin films; reactive magnetron sputtering; electron transport layer

## 1. Introduction

Perovskite solar cells (PSCs) are developing rapidly, and the power conversion efficiency (PCE) of these devices have increased from 3.8% [1] to over 20% [2] in a very short time, and continues to be optimized [3]. In the past, mesoporous perovskite has always been a classical structure with high efficiency [4]. At present, the efficiency of the device based on this classic structure has reached 23.2% [3]. Titanium oxide ($TiO_2$) is a widely-used electron transport material in perovskite solar cells, and mesoporous layer $TiO_2$ is the most typical nanostructure. The dense $TiO_2$ acts to transport electron blocking holes, and the function of the porous $TiO_2$ is to improve the uniformity of the spin-coated perovskite film. However, since the preparation of titanium dioxide needs to undergo a high temperature process, has weak optical stability and low mobility, leading to higher production costs and longer energy recovery times [5], its development is greatly limited [6]. On this basis, the fabrication of planar PSCs and electron transport layers has attracted extensive research [7]. Developing new electron transport layer materials, such as $Zn_2SnO_4$ [8], $In_2O_3$ [9], $WO_x$ [10], ZnO [11], PCBM [12],

and $SnO_2$ [13] is a way to solve the shortcomings of titanium oxide. Among these materials, the most promising substitution for $TiO_2$ is $SnO_2$, because $SnO_2$ does not require high temperature preparation and has excellent photoelectric properties such as high electron mobility, suitable band gap and band structure, and high transparency [13]. The initial studies of $SnO_2$ in PSCs were almost concurrently carried out in many research teams, such as Dai and co-workers and the Guojia Fang teams [13], who reported an efficiency of ~3.9% and over 20% [14], respectively. After that, $SnO_2$ ETLs attracted more and more attention and made rapid progress [6]. At present, the device with $SnO_2$ as the ETLs has reached 21.52% efficiency [15], although there is still great potential for further increasing that efficiency [16].

Many methods were attempted to synthesize tin oxide [13]. Ke et al. used air annealing of $SnO_2$ film at 180 °C, made by the low-temperature solution method for 1 h, to realize 17.21% efficiency in planar perovskite [17]. Hagfeldt and co-workers deposited $SnO_2$ layers by ALD (atomic layer deposition) technology applied to planar PSCs with an efficiency over 18% and a high voltage of 1.19 V [18]. In addition, Hagfeldt and co-workers used chemical bath deposition (CBD) and spin coating to obtain $SnO_2$ layers, with the efficiency of the device made by this method being more than 20.7% and also having better stability performance [13]. Jen and co-workers prepared $SnO_2$ as an electron transport layer by electrochemical deposition and applied it to planar PSCs with an efficiency of 13.88% [19]. The latter has the advantages of low temperature treatment and no annealing treatment. In summary, the solution method is the main method to prepare $SnO_2$ as an ETL, and the PSCs production efficiency is higher than other methods, but spin coating and annealing limit the application of PSCs commercialization [20]. Moreover, the low deposition rate, being time consuming and high cost of making tin oxide electron transport layer by ALD technology also limit its application for industrialization [21]. On the contrary, radio frequency magnetron sputtering (RS) is a mature technology for low cost industrialization of semiconductor materials [6,22]. RS has played an important role in making electron transport and hole barrier layers in PSCs [23], including improving device performance [24], lowering temperature and reducing costs [25], as well as providing a basis for large-scale and industrialized solar cells that cannot be replaced by other methods [26]. There are a few reports about the application of $SnO_2$ electron transport layer made by magnetron sputtering to planar PSCs [6]. Giulio et al. has prepared $SnO_2$ film by reactive sputtering for application in gas sensors [27]. Qiu et al. used tin oxide prepared by sputtering and obtained very good results in perovskites [28]. In addition, reactive magnetron sputtering can prepare a compound film of a compound ratio by adjusting parameters [29], thereby realizing the controllability of the film and having less restrictions on the substrate [30], and depositing $SnO_2$ on various substrates [30] which is very suitable for large-area uniform coating and industrial production [31]. The radio frequency (RF) sputter-deposited film is relatively denser and flatter, but we should not easily conclude before further studying the two sputtering methods. In addition, since the tin target is a pure metal target, the target is easier to prepare due to the ductility of the metal during preparation, compared to the tin oxide target. Moreover, the tin target requires less excitation energy at the time of sputtering than the tin oxide target because the molecular weight of tin is smaller than the molecular weight of tin oxide. In this paper, the preparation of $SnO_2$ thin film electron transport layer by radio frequency magnetron reactive sputtering at room temperature and its application to planar PSCs was studied. It has been found through research that $SnO_2$ film prepared at room temperature has a smooth surface, good transmittance, a wide band gap, and a suitable energy level position, which makes $SnO_2$ film sputtered at room temperature have good electron transport layer properties. Finally, $SnO_2$ is applied in planar PSCs to obtain an efficiency of 14.63%. This paper provides a simpler and more cost-effective method for the application of $SnO_2$ to PSCs, which is conducive to the promotion and application of large areas.

## 2. Experimental Section

### 2.1. Preparation of SnO$_2$ Film

At room temperature, SnO$_2$ thin films prepared by radio frequency reactive magnetron sputtering were deposited on FTO (SnO$_2$:F). FTO layers on glass substrates were cleaned with alkaline detergent, acetone, absolute ethanol and deionized water for 15 min. The cleaned substrates were then UV–ozone treated for 15 min before they were used for sputtering. A 99.99% pure tin (Sn) target (the target size is Φ 50 × 4 mm) was fixed in the vacuum chamber and the FTO glass was fixed on the substrate, with the height between Sn target and FTO substrate being close to 50 cm. The base pressure was pumped to $6 \times 10^{-4}$ Pa before any sputtering was carried out. The incoming gas consisted of 99.99% pure argon (Ar) and 99.99% pure oxygen (O$_2$). The flow rate of argon and oxygen was 9 to 1. The surface impurities of the FTO glass substrate were cleaned by intermediate frequency cleaning for 10 min, and the target surface impurities were removed by 15 min pre-sputtering before sputtering. The sputtering power of the Sn target was 50 W with the sputtering time being set to 10, 20 and 30 min (marked as sample RS–10, RS–20, and RS–30). When starting to sputter the tin target, the operating frequency was 13.56 MHz and the waveform was a Sine wave. In addition, the substrate bias was 180 V.

### 2.2. Fabrication of Planar Perovskite Solar Cells

Using a blank FTO (SnO$_2$:F) glass substrate as a comparison, a perovskite layer was deposited by a one-step method, and a prepared perovskite solution (CH$_3$NH$_3$I and PbI$_2$ = 1:1) was applied to the sample at a rotation speed of 3000 rpm for 30 s [32,33]. The perovskite films were annealed at 100 °C for 10 min. After cooling to room temperature, 72.3 mg Spiro–OMeTAD, 28.8 μL 4–tert–butylpyridine and 17.5 μL of Li–bis (trifluorometha–nesulfonyl) imide solution (520 mg·mL$^{-1}$ in acetonitrile) were dissolved in 1 mL chlorobenzene to form a Spiro–OMeTAD solution, and the Spiro–OMeTAD layer was sequentially spin-coated at 3000 rpm for 30 s so as to form the hole transfer layer. The gold electrode was then deposited on the Spiro–OMeTAD layer by vacuum evaporation to help form the entire perovskite device with an active area of 0.1 cm$^2$. The structure of the whole device is shown in Figure 1a.

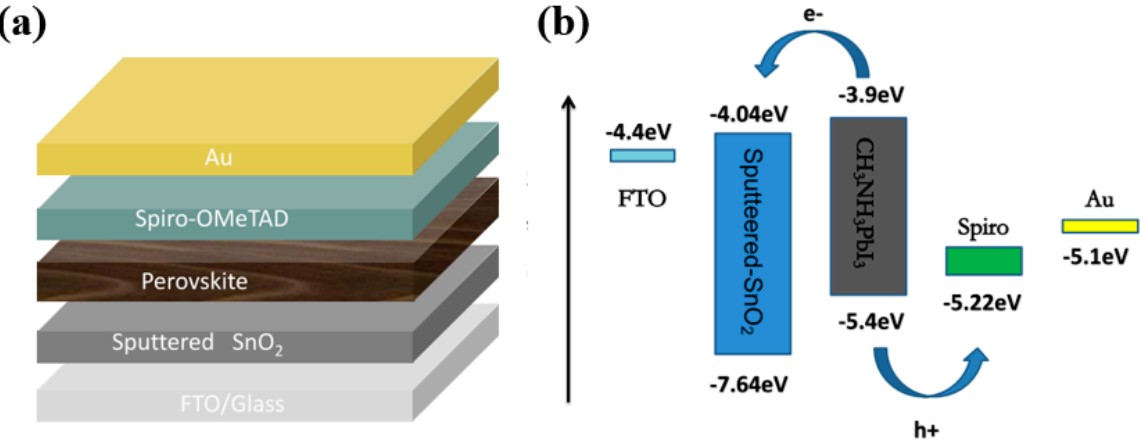

**Figure 1.** (**a**) Structure diagram and (**b**) band diagram of the device.

### 2.3. Characterization

Field-emission scanning electron microscopy (FESEM) images were obtained using a ZEISS SUPRA55 (Oberkochen, Germany). AFM figures were measured using a 300 HV scanning force microscope (SEIKO, Tokyo, Japan). The photovoltaic performance of PSCs was recorded using a Keithley 4200 source meter under a one-sun AM 1.5G (100 mW·cm$^{-2}$) illumination with a solar light simulator (Newport Oriel Sol3A Class AAA, 64023A Simulator, Mahwah, NJ, USA), which

was calibrated using an NREL standard Si solar cell. The UV–Vis light absorption measurement was performed by using an ultraviolet-visible (UV–Vis) spectropho-tometer (Shimadzu UV-3101 PC, Kyoto, Japan). X-ray photoelectron spectroscopy (XPS) and ultraviolet photoelectron spectrum (UPS) measurements were conducted using an ESCALAB 250Xi (Thermo, Waltham, MA, USA) system. For XPS, the relevant parameters when measuring were as follows: Monochrome Al K$\alpha$ ($h_v$ = 1486.6 eV), power of 150 W, a 500 μm beam spot, a charge correction using contaminated carbon (C 1$s$ = 284.8 eV) for correction, a constant analyzer pass energy ($E_p$), an energy narrow scan of 20 eV, and a vacuum of $1 \times 10^{-10}$ mba. External quantum efficiency (EQE) measurements were obtained on a Keithley 2000 multimeter (Cleveland, OH, USA) as a function of the wavelength from 350 to 800 nm on the basis of a Spectral Products DK240 monochromator. The PL spectra and fluorescence decay curves were taken out with a combined steady state fluorescence spectrometer (FLS980, Edinburgh, UK). The active area of the cell was 0.1 cm$^2$. All samples were measured in air (25 °C).

## 3. Results and Discussion

The perovskite layer requires good absorption of light. It can be seen from the transmission spectrum in Figure 2a that the sputtered SnO$_2$ film exhibited high light transmittance. In addition, the band gap of the sputtered SnO$_2$ film was calculated from the transmission spectrum to be 3.79 eV, which is shown in the inset of Figure 2. The work function ($\phi$) can be determined by the difference between the photon energy and the binding energy of the secondary cutoff edge. Combined with Figure 2b, we can get the conduction band and Fermi level. The position of the valence band was −7.83 eV, respectively, and the position of the conduction band was −4.04 eV. In addition, the Fermi level was −4.74 eV. Figure 1b shows the band diagram of a planar perovskite with SnO$_2$ prepared by reactive sputtering as an electron transport layer. It can be seen that the SnO$_2$ band structure prepared by reactive sputtering can completely transport the electron and hole properties of planar perovskite solar cells. Moreover, the valence band of the SnO$_2$ prepared by reactive sputtering is lowered by −5.4 eV. This shows that the sputtered tin oxide is suitable for the electron transport layer. The survey of XPS spectra in Figure 3a,b confirms the presence of Sn and O in reactive sputtered SnO$_2$ film [34]. Compared with other literature, it can be seen that the Sn peak shifts, indicating the presence of oxygen defects in the reactive sputtered SnO$_2$ [17]. The binding energies of the Sn 3$d_{5/2}$ and Sn 3$d_{3/2}$ peaks in the literature are 487.11 and 495.56 eV. However, the binding energies of tin oxide prepared by reactive sputtering are 495.14 and 486.69 eV, respectively. The tin oxide binding energy prepared by reactive sputtering is 0.42 eV lower than that found in the literature. It shows that the tin oxide prepared by reactive sputtering contains a part of stannous fluoride, which is not completely tin oxide. In addition, we performed an EDS on the tin oxide layer and found that Sn:O is 1:2. This shows that SnO$_2$ prepared by reactive sputtering also has great potential. However, it can still be drawn that reactive sputtering of SnO$_2$ is very suitable for the electron transport layer in planar.

The roughness of SnO$_2$ prepared by reactive sputtering was detected by AFM, and blank FTO glass was used as a comparative sample. The measured value of the roughness is the average value of the three different areas on the surface of the sample. As shown in Figure 4a,b, the FTO glass without sputtering has a roughness of 31.8 nm and the roughness of the sample with sputtering is 30.2 nm. It can be seen that the surface of the FTO glass has a lower roughness after sputtering. It can be inferred that the re-filling of the sputtered SnO$_2$ nanoparticles onto the surface of the FTO glass reduces the surface roughness of the sample. In order to investigate the effect of this roughness variation on the perovskite layer, the surface morphology of the perovskite layer was observed on different substrates by FESEM. Figure 4c,d shows the FESEM morphology of a perovskite film deposited on the surface of FTO glass both with unsputtered and sputtered SnO$_2$. It is obvious that the film deposited on the bare FTO by the perovskite film is rough and contains many large grains. However, the perovskite layer exhibits a flatter surface in the film of SnO$_2$ prepared by reactive sputtering, and the grain size is more uniform with almost no large grain size. It can be concluded that the SnO$_2$ film prepared by reactive sputtering has a positive effect on the subsequent perovskite layer deposition. Since one of the

keys to a well-prepared perovskite solar cell is the interface, a good interface means a good device. On the one hand, reactive sputtering of the $SnO_2$ film on the FTO surface effectively improves the surface, and the surface of the FTO is flatter, which provides a favorable position for the crystallization of the perovskite grains, making perovskite film smoother and more uniform. Therefore, $SnO_2$ film prepared by reactive sputtering is very suitable for the electron transport layer material, as it effectively improves the crystallization of the perovskite layer, makes the perovskite smaller in size, and has a smoother surface, which reduces defects in interface contact. This kind of change is beneficial to improve the performance of the perovskite solar cell, which will be further proved by the photoelectric parameters of the device.

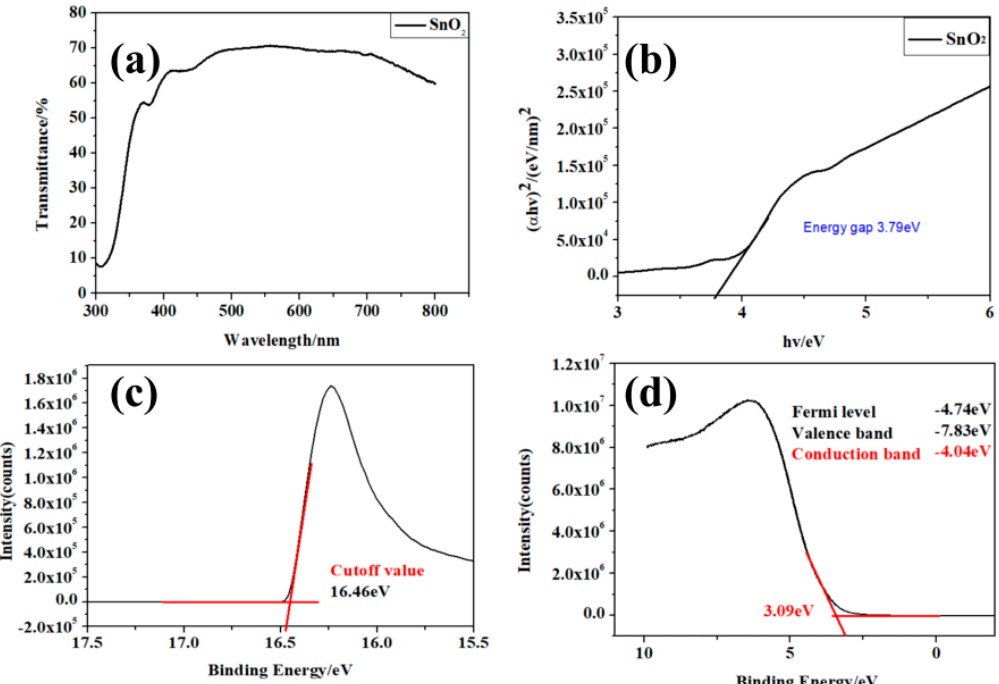

**Figure 2.** (**a**,**b**) Transmittance spectrum and Tauc plots of the $SnO_2$ film sputtered on glass. (**c**,**d**) Ultraviolet photoelectron spectrum (UPS) of $SnO_2$ film sputtered on silicon wafer.

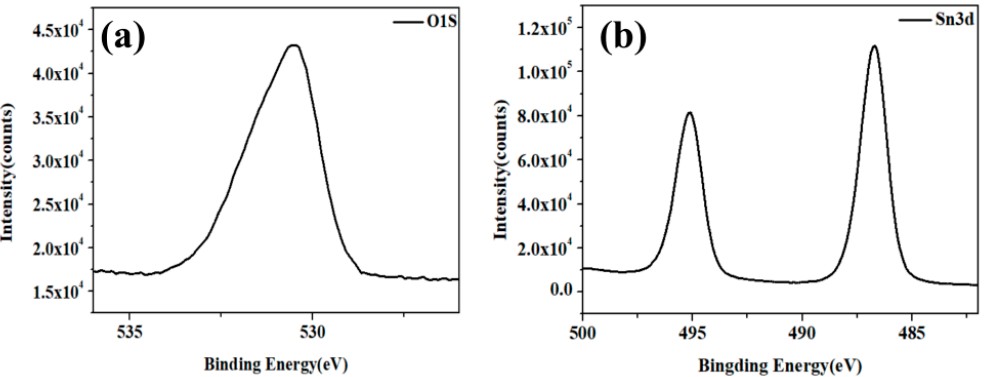

**Figure 3.** (**a**,**b**) The survey of XPS spectra of the $SnO_2$ film on the FTO substrate.

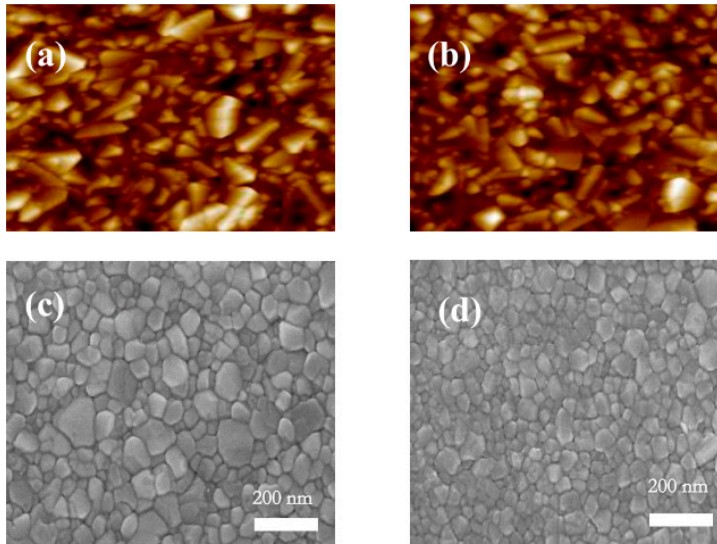

**Figure 4.** AFM images of surface morphology of FTO substrate without and with sputtered SnO$_2$ film (**a**,**b**), and SEM images of perovskite films deposited on FTO substrate without and with reactive-sputtered SnO$_2$ film (**c**,**d**).

The electron transport layer of planar perovskites, the deposition rate and sputtering time of SnO$_2$ film prepared by reactive sputtering have a crucial influence on the optical and electrical properties of the device. SnO$_2$ was deposited on FTO for 50 min, and the thickness was determined by electron microscopy to be 30.5 nm, thereby confirming the deposition rate of 0.61 nm/min. The thickness of the deposit varies linearly. Therefore, it is known that the deposition rate of SnO$_2$ prepared by reactive sputtering is 0.61 nm/min. As a result, the thickness of sample RS–10, sample RS–20, and sample RS–30 was about 6.1, 12.2 and 18.3 nm, respectively. Moreover, the effect of reactive sputtering time on the photoelectric properties of planar perovskites and the photoluminescence spectra of the corresponding samples were also investigated. The results are shown in Figure 5 and Table 1. Planar perovskite cells based on bare FTO have an efficiency of only 5.08% due to a poor short-circuit current ($J_{sc}$) and fill factor (FF). After using SnO$_2$ prepared by reactive sputtering as an electron transport layer, the performance of the entire device was greatly improved.

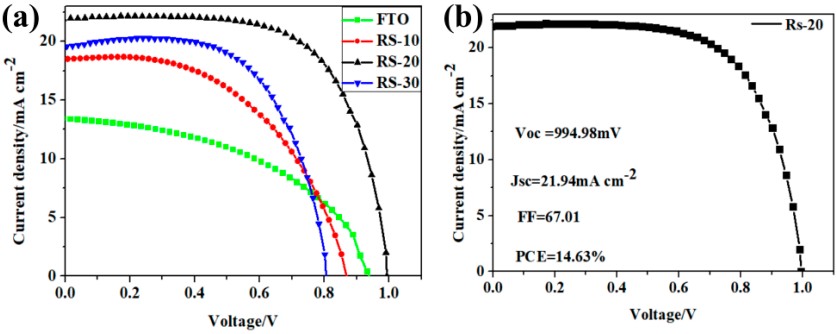

**Figure 5.** *J–V* average curves of different planar perovskite solar cells (PSCs) (**a**,**b**) *J–V* curve of the best planar PSC based on room temperature reactive sputtering of SnO$_2$.

**Table 1.** Average photoelectric parameters of PSCs with different reaction sputtering times.

| Sample | Sputtering Time (min) | Voc/V | $J_{sc}$/mA·cm$^{-2}$ | FF (%) | PCE (%) |
|--------|----------------------|-------|----------------------|--------|---------|
| FTO    | 0                    | 0.99  | 10.31                | 49     | 5.08    |
| RS–10  | 10                   | 0.93  | 15.91                | 65     | 9.56    |
| RS–20  | 20                   | 0.95  | 20.20                | 62     | 12.02   |
| RS–30  | 30                   | 0.86  | 20.21                | 56     | 9.70    |

The efficiency of the device reaches 12.02%, the current reaches 20.20 mA·cm$^{-2}$, and the filling factor reaches more than 65. On the one hand, SnO$_2$ prepared by reactive sputtering has a suitable band gap and energy level, which promotes electron transport and blocks holes. On the other hand, the perovskite layer becomes smoother on the SnO$_2$ layer prepared by reactive sputtering compared to the bare FTO, improving $J_{sc}$ and FF.

In contrast, the best reactive sputtering time is 20 min, and the sample RS–20 shows the best device performance. The FF of sample RS–10 was much larger than the FF of sample FTO, but it is very close to the FF of sample RS–20, indicating that the length of reactive sputtering will have an impact on the coverage of SnO$_2$ on the FTO surface. The longer the reactive sputtering time, the higher the coverage rate until the full coverage increases the thickness of SnO$_2$ as the electron transport layer. When the sputtering time exceeds 20 min, the tin oxide gradually becomes thicker, the resistance of the electron transport layer is increased, electrons are not efficiently transported, carrier recombination is increased, and the device performance deteriorates. $J_{sc}$ first increases with the increase of sputtering time and then remains stable for a certain period of time, but as the sputtering time increases, $V_{oc}$ remains stable and then decreases within a certain range. It can be inferred that FF and $J_{sc}$ keep increasing with the increase of SnO$_2$ coverage. After complete coverage, the internal resistance increases as the thickness of SnO$_2$ increase, but Voc decreases. As a result, 20 min is the optimum reaction sputtering time. In addition, it can be seen from the photoluminescence spectrum that when the perovskite layer is directly deposited on the surface of the FTO glass, it has the highest PL peak intensity, which indicates that the carrier recombination is the most serious. However, when SnO$_2$ film of RS–10 is added, the peak intensity is found to be significantly weakened. Therefore, SnO$_2$ can effectively suppress the recombination of carriers. However, due to the short time, the thinner thickness may not completely cover the FTO layer, which results in the peak strength to not be minimal. When the time reaches 20 min, the peak intensity is minimized, with the smallest carrier recombination as well as the highest photoelectric conversion efficiency being achieved. As time continues to 30 min, it can be seen that the PL peak will become larger, which indicates that the charge recombination is also increasing, thereby affecting the performance of the entire device. Therefore, the optimal reaction sputtering time is 20 min.

It can be seen that the main photoelectric properties of the reactive-sputtered SnO$_2$ of the planar perovskite are short-circuit current and fill factor. Therefore, we have studied the external quantum efficiency (EQE) of a planar perovskite with SnO$_2$ as the electron transport layer and with no SnO$_2$, as shown in Figure 6a. The $J_{sc}$ of sample RS–20 was calculated from the EQE curve to be 20.07 mA·cm$^{-2}$, which is close to the experiment value. It can also be seen that the SnO$_2$ ETL prepared by reactive sputtering effectively improves the EQE curve intensity and range of the PSCs. The reactive sputter-deposited SnO$_2$ has an impact on the EQE performance of the device in two aspects. On the one hand, the deposition of SnO$_2$ ETLs improves the deposition of the perovskite layer as well as the light absorption performance of the device. On the other hand, reactive sputter-deposited SnO$_2$ acts as an interfacial layer between the perovskite layer and the FTO glass substrate, thereby greatly improving the carrier transport and hole-blocking properties of the device. It can be seen from Figure 7c that the carrier transport of the perovskite film containing the tin oxide layer is faster than that of the perovskite layer without the tin oxide layer, further demonstrating that the tin oxide prepared by reactive sputtering is suitable as the electron transport layer. This is consistent with the results of PL. In addition, the CV curve of the dark state CV curve and the illumination condition were also consistent. We first examined the CV curve, then the dark CV curve, which may have some effect.

This proves that the SnO$_2$ film prepared by reactive sputtering at room temperature is feasible as an ETL of a perovskite device. In addition, the optimal deposition efficiency of the perovskite layer was optimized by optimizing the reactive sputtering conditions, and the best photoelectric efficiency was obtained in sample RS–20, as shown in Figure 5b. The parameters are as follows: $V_{oc}$ = 994.98 mV, $Jsc$ = 21.94 mA·cm$^{-2}$, FF = 67.01, and PCE = 14.63%. In Figure 8, we show a histogram of the performance parameters of 30 devices, which shows that we have good repeatability.

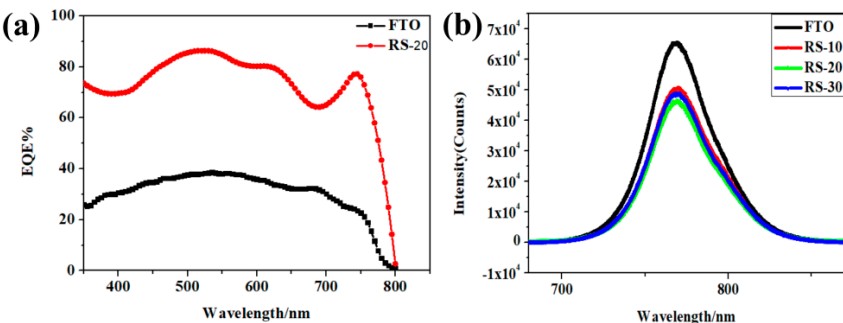

**Figure 6.** (**a**) External quantum efficiency (EQE) curves of different planar PSCs and (**b**) the corresponding photoluminescence spectra of the samples.

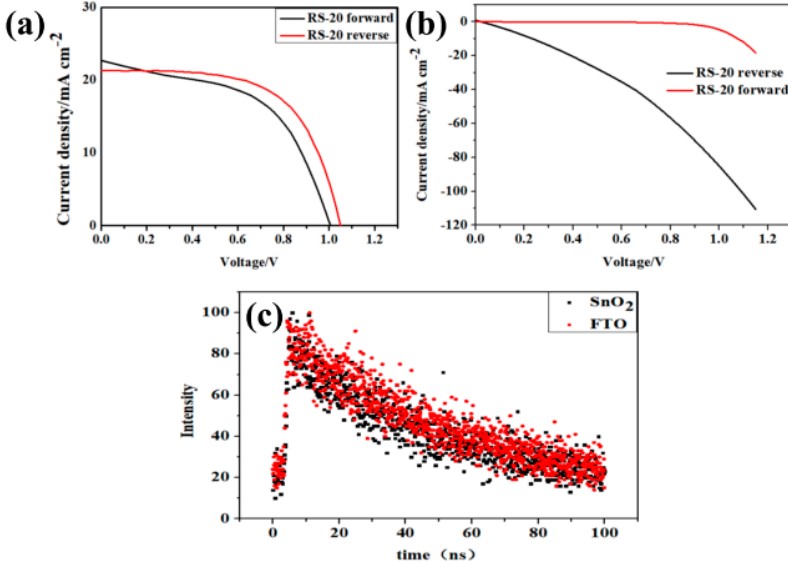

**Figure 7.** (**a**) Forward and reverse bias *J–V* characteristics of the RS–20; (**b**) *J–V* characteristics of the RS–20 under dark illumination; (**c**) Time resolved photoluminescence (TRPL) curve of the FTO substrate and the sputtered tin oxide substrate.

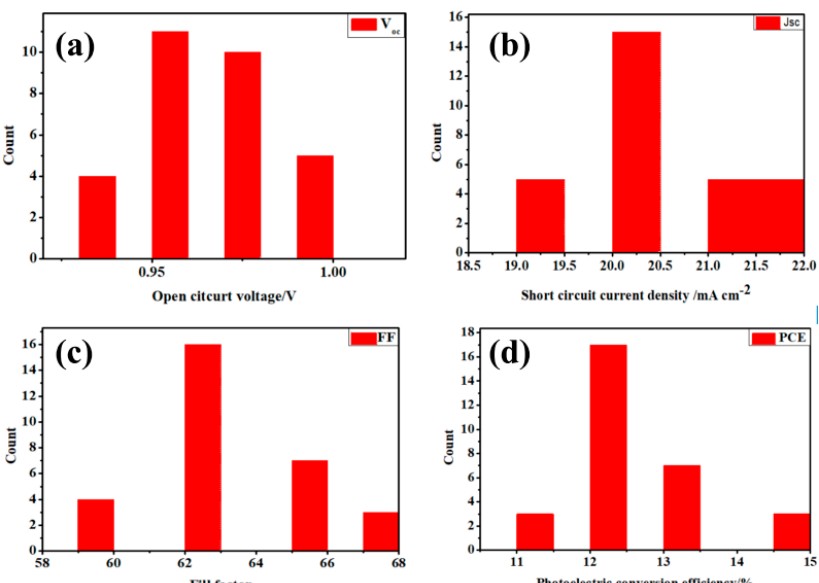

**Figure 8.** Histograms of (**a**) Voc, (**b**) Jsc, (**c**) FF, and (**d**) PCE of 30 PSCs based on $SnO_2$ HBL reactive sputtering.

## 4. Conclusions

In general, this work illustrates the potential of reactive sputtering to produce $SnO_2$ as a planar perovskite electron transport layer at room temperature. $SnO_2$ film prepared by reactive sputtering has a high transmittance and a suitable energy band structure. The ultra-thin $SnO_2$ improves the roughness of the perovskite layer, thereby effectively strengthening the planar perovskite into electrons and air. In addition, the reactive sputtering preparation of $SnO_2$ has good repeatability. Moreover, the highest efficiency of perovskite prepared by reactive sputtering based on $SnO_2$ reached 14.63%, which will open new ideas for large-area applications and commercial applications of perovskites. This is far from the limit of reactive tin oxide preparation, and various parameters have yet to be improved, thereby helping to improve the application capacity of perovskites.

**Author Contributions:** Conceptualization, W.S. and R.M.; Methodology, W.S.; Software, S.W., S.L., Y.Z., X.M., and C.D.; Formal Analysis, S.W.; Investigation, W.S.; Resources, C.W.; Data Curation, W.S.; Writing—Original Draft Preparation, W.S.; Writing—Review and Editing, W.S.; Visualization, S.W.; Supervision, R.M.; Project Administration, C.W.; Funding Acquisition, C.W.

**Funding:** This research was funded by the Fundamental Research Funds for the Central Universities, No. 230201606500078. This research was funded by the National Natural Science Foundation of China, Nos. U1302274 and 51674026.

**Conflicts of Interest:** The authors declare no conflict of interest.

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
