# Peer review of "Reactive-Sputtered Prepared Tin Oxide Thin Film as an Electron Transport Layer for Planar Perovskite Solar Cells"

_coatings, doi:10.3390/coatings9050320_

Reviewer 1 Report

Comments to the authors

The manuscript entitled “Reactive sputtering tin oxide thin film as electron transport layer for planar perovskite solar cells” investigates use of SnO2 deposited by reactive sputtering as an electron transport layer (ETL) for planar perovskite solar cells. The work is interesting, however, is lacking quality in terms of presentation. It can be considered for publication in “Coatings”, if the authors address the following comments.

1.      The conclusion states “In addition, the reactive sputtering preparation of SnO2 has good repeatability.”. There is no repeatability data to support this statement.

2.      The transmittance of the SnO2 film is ~ 70%, which is lower than FTO glass (~ 80%). This decrease in the transmittance is quite significant for an ETL layer for perovskite solar cell application. Please comment.

3.      For the photovoltaic performance parameters of the perovskite solar cells, statistical variation of the cell performance should be provided (average values and standard deviation). Need to state how many devices were prepared for evaluating each condition. Presenting a histogram would be preferred.  

4.      The conclusion states “The ultra-thin SnO2 improves the roughness of the perovskite layer, thereby effectively strengthening the planar perovskite into electrons and air.” What does the latter half of this statement mean?

5.      The Voc values provided in Table 1 are misrepresented. Double check.

6.      The Voc and Jsc values in Table 1 do not match with J-V curve in Fig. 5a. The paper does not indicate why are these values different. Is it the average values presented in the table and the ones in the J-V curves present the best performing cells for each condition?

7.      14.63% is lower compared to other reports on SnO2 based planar perovskite solar cells (See review paper Adv. Funct. Mater. 2018, 1802757). The authors should put forward their outlook on how the efficiency can be further improved to make this work more appealing.

8.      Several typos can be found, which affects the readability of the work. (e.g. figure numbers in the text are not correct, line 190, line 233 etc.

Author Response

Dear Editor:

Many thanks to the reviewers for the insightful comments and suggestions concerning our manuscript, titled “Reactive sputtering tin oxide thin film as electron transport layer for planar perovskite solar cells” We have studied the comments carefully and have made corrections. Hopefully, these changes will make the manuscript more acceptable for publication.

Responds to the reviewer’s comments:

Review #1:

(1)   The conclusion states “In addition, the reactive sputtering preparation of SnO2 has good repeatability.” There is no repeatability data to support this statement.

Response: We are very sorry for our negligence. Because magnetron sputtering is very mature, we have not shown it in the text, but we have added a histogram of the efficiency distribution to illustrate this aspect in Figure 8.

(2)   The transmittance of the SnO2 film is ~70%, which is lower than FTO glass (~80%). This decrease in the transmittance is quite significant for an ETL layer for perovskite solar cell application. Please comment.

Response: Thanks for the comment. Since the FTO glass produced by different manufacturers has different transmittance, electrical conductivity and electrical resistance, it is normal when our FTO glass plus tin oxide electron transport layer is different.

(3)   For the photovoltaic performance parameters of the perovskite solar cells, statistical variation of the cell performance should be provided (average values and standard deviation). Need to state how many devices were prepared for evaluating each condition. Presenting a histogram would be preferred.

Response: We presented the preparation processes in Figure 8.

(4)   The conclusion states “The ultra-thin SnO2 improves the roughness of the perovskite layer, thereby effectively strengthening the planar perovskite into electrons and air.” What does the latter half of this statement mean?

Response: We are very sorry for our negligence. Ultra-thin SnO2 improves the roughness of the perovskite layer, effectively improving the ability of planar perovskites to transport electrons and holes.

(5)   The Voc values provided in Table 1 are misrepresented. Double check.

Response: This is our writing mistake, thank you for pointing out our shortcomings. It has been corrected. We hope that these revisions are satisfactory.

(6)   The Voc and Jsc values in Table 1 do not match with J-V curve in Fig. 5a. The paper does not indicate why are these values different. Is it the average values presented in the table and the ones in the J-V curves present the best performing cells for each condition?

Response: Thanks for the comment. As you said, the values in the table are average values, and the values in the J-V curve are the best values under this condition. It has been corrected in the manuscript .

(7)   14.63% is lower compared to other reports on SnO2 based planar perovskite solar cells (See review paper Adv. Funct. Mater. 2018, 1802757). The authors should put forward their outlook on how the efficiency can be further improved to make this work more appealing.

Response: Thanks for the comment. Our low efficiency is related to our environment. Different temperatures and humidity will affect our efficiency. This article has achieved good results in the best sputtering conditions using tin oxide target sputtering, but there are still many shortcomings in direct sputtering with tin targets and different sputtering methods. Nickel oxide and titanium oxide have been explored to directly distinguish between metal targets and different sputtering methods, and tin oxide has great research potential.

(8)   Several typos can be found, which affects the readability of the work. (e.g., figure numbers in the text are not correct, line 190, line 233 etc.

Response: Thanks for the comment. We have corrected in the manuscript.

We tried our best to improve the manuscript and made some changes in the manuscript. These changes will not influence the content and framework of the paper. We did not list the changes but marked in red font in revised paper.

Once again, thank you very much for your comments and suggestions.

Reviewer 2 Report

Please, find the attached file.

Reviewer 3 Report

Comments to the authors:

Sun et al. are reporting on using reactive sputtering for Tin Oxide as ETL in perovskite solar cells. The deposition was done at room temperature, aims for lower cost as lower energy is required as well as large scale industrialization. The research that is presented here addresses important points for perovskite solar cells. However, the work seems to be a replication of work published few months ago by Qiu et al. [Scalable Fabrication of Stable High Efficiency Perovskite Solar Cells and Modules Utilizing Room Temperature Sputtered SnO2 Electron Transport Layer, Adv. Funct. Mater. 2018, 1806779]. The authors might not be aware of this. The authors claim in page 2, line 69, that none of the published work was prepared by reactive sputtering “…none of them is made by reactive sputtering.”, which is not true as presented by Qiu et al. in the provided reference.

The authors in the abstract, conclusion and in page 2 line 74-81, states their findings using the reactive sputtering at room temperature, where they achieved smooth surface (RMS 30nm), wide band gap, suitable energy levels, and 14.6% PCE. They used XPS, UPS, uv-vis, SEM, AFM, and solar cell devices. Qiu et al. used same method of reactive sputtering at room temperature for preparing SnO2 ETL in planar structure and characterizing with same methods and some extra. Qiu achieved substantially better PCE of 20% and also tested for 5x125px area devices with 12%PCE, better transmittance, similar band alignment, controlled and better surface roughness of 25 nm. Indeed Qiu et al. used different perovskites with mixed cations but the material under investigation in both of these studies is the ETL for perovskite solar cells.

I do not see anything new in this study or any addition to what was published. I would recommend changing the focus of this work by studying different aspects of reactive sputtering for perovskite solar cells, which includes complete change of the article and probably different submission.

Author Response

Dear Editor:

Many thanks to the reviewers for the insightful comments and suggestions concerning our manuscript, titled “Reactive sputtering tin oxide thin film as electron transport layer for planar perovskite solar cells” We have studied the comments carefully and have made corrections. Hopefully, these changes will make the manuscript more acceptable for publication.

Responds to the reviewer’s comments:

Review #3:

  Sun et al. are reporting on using reactive sputtering for Tin Oxide as ETL in perovskite solar cells. The deposition was done at room temperature, aims for lower cost as lower energy is required as well as large scale industrialization. The research that is presented here addresses important points for perovskite solar cells. However, the work seems to be a replication of work published few months ago by Qiu et al. [Scalable Fabrication of Stable High Efficiency Perovskite Solar Cells and Modules Utilizing Room Temperature Sputtered SnO2 Electron Transport Layer, Adv. Funct. Mater. 2018, 1806779]. The authors might not be aware of this. The authors claim in page 2, line 69, that none of the published work was prepared by reactive sputtering “…none of them is made by reactive sputtering.”, which is not true as presented by Qiu et al. in the provided reference.

The authors in the abstract, conclusion and in page 2 line 74-81, states their findings using the reactive sputtering at room temperature, where they achieved smooth surface (RMS 30nm), wide band gap, suitable energy levels, and 14.6% PCE. They used XPS, UPS, uv-vis, SEM, AFM, and solar cell devices. Qiu et al. used same method of reactive sputtering at room temperature for preparing SnO2 ETL in planar structure and characterizing with same methods and some extra. Qiu achieved substantially better PCE of 20% and also tested for 5x125px area devices with 12%PCE, better transmittance, similar band alignment, controlled and better surface roughness of 25 nm. Indeed Qiu et al. used different perovskites with mixed cations but the material under investigation in both of these studies is the ETL for perovskite solar cells.

I do not see anything new in this study or any addition to what was published. I would recommend changing the focus of this work by studying different aspects of reactive sputtering for perovskite solar cells, which includes complete change of the article and probably different submission.

 Response: Thank you for the comment. First of all, our work is earlier than the work of Qiu et al., and I have not copied their work, just want to study more carefully.
Secondly, our work is very different from them. First, our target is tin target, they use tin oxide target, and second, we use RF power.The effects of different sputtering power supplies and targets are very different.Moreover, many people in nickel oxide and titanium oxide have studied with DC power source and RF power source and nickel-titanium oxide target nickel target titanium target to find the best sputtering conditions and methods. Specific reference can be made to this document (   Yan X, Zheng J, Zheng L, Lin G, Lin H, Chen G, Du B, Zhang F. Optimization of sputtering NiO x films for perovskite solar cell applications. MATER RES BULL 2018;103:150) This is insufficient in tin oxide. Therefore, our work and the work of Qiu et al. are very different.

We tried our best to improve the manuscript and made some changes in the manuscript. These changes will not influence the content and framework of the paper. We did not list the changes but marked in red font in revised paper.

      Once again, thank you very much for your comments and suggestions. 

Round  2

Reviewer 1 Report

Comment to authors

The authors have answered most of the questions adequately, however, the authors should further address the following two points to make this work acceptable for publication.

1.      The response to comment (7) is not adequate. Based on the authors response, it implies that 14.63% is the ultimate efficiency possible using their approach. Is this true? If it is, the conclusion would be that reactive sputtering of SnO2 in perovskite solar cell is not a promising approach. If it is not, the conclusion should put the future works that the authors plan to input for further optimization of this work.

2.      In the revised manuscript, the authors have put the TRPL response of the perovskite solar cells (Figure 7c). But no proper explanation with respect to the figure is provided. The authors should tell the reader on how to interpret the figure, which is very important.

Author Response

Dear Editor:

Many thanks to the reviewers for the insightful comments and suggestions concerning our manuscript, titled “Reactive sputtering tin oxide thin film as electron transport layer for planar perovskite solar cells” We have studied the comments carefully and have made corrections. Hopefully, these changes will make the manuscript more acceptable for publication.

Responds to the reviewer’s comments:

Review #1:

1    The response to comment (7) is not adequate. Based on the authors response, it implies that 14.63% is the ultimate efficiency possible using their approach. Is this true? If it is, the conclusion would be that reactive sputtering of SnO2 in perovskite solar cell is not a promising approach. If it is not, the conclusion should put the future works that the authors plan to input for further optimization of this work.

Response: Thanks for the comment. Your opinion is very useful to us, which we have not explained. 14.6% is by no means the limit of reactive sputtering. We have been pursuing higher efficiency because there is also a lot of work that we need to do. For example, the best sputtering pressure, the best substrate rotation rate, the best sputtering power, the optimum substrate temperature, we just explored that this is a feasible path, and it seems that this path is feasible. But still need more work to study. We have already explained the future work in the conclusion, expecting more people to pay attention and study together. We hope that these revisions are satisfactory.

2    In the revised manuscript, the authors have put the TRPL response of the perovskite solar cells (Figure 7c). But no proper explanation with respect to the figure is provided. The authors should tell the reader on how to interpret the figure, which is very important.

Response: Thanks for the comment. It can be seen from Fig. 7c that the carrier transport of the perovskite film containing the tin oxide layer is faster than that of the perovskite layer without the tin oxide layer, further demonstrating that the tin oxide prepared by reactive sputtering is suitable as the electron transport layer. This is consistent with the results of PL.

We tried our best to improve the manuscript and made some changes in the manuscript. These changes will not influence the content and framework of the paper. We did not list the changes but marked in red font in revised paper.

Once again, thank you very much for your comments and suggestions.

Reviewer 2 Report

The revised manuscript looks good though it has some typos and Figure 7c needs to be improved for its graphic quality. In line 99, correct the subscript for SnO2:F and also In Figure 2 (b), in y-axis, I think, it should be  symbol α instead of a.

I recommend to accept this manuscript after correcting these minor errors. 

Author Response

Dear Editor:

Many thanks to the reviewers for the insightful comments and suggestions concerning our manuscript, titled “Reactive sputtering tin oxide thin film as electron transport layer for planar perovskite solar cells” We have studied the comments carefully and have made corrections. Hopefully, these changes will make the manuscript more acceptable for publication.

Responds to the reviewer’s comments:

Review #2:

(1)    The revised manuscript looks good though it has some typos and Figure 7(c) needs to be improved for its graphic quality. In line 99, correct the subscript for SnO2:F and also In Figure 2 (b), in y-axis, I think, it should be  symbol α instead of a.

Response: Thanks for the comment.

SnO2:F               SnO2:F

We have corrected it in the manuscript. We hope that these revisions are satisfactory.

Images cannot be uploaded and has been modified in the manuscript

We tried our best to improve the manuscript and made some changes in the manuscript. These changes will not influence the content and framework of the paper. We did not list the changes but marked in red font in revised paper.

Once again, thank you very much for your comments and suggestions.

Reviewer 3 Report

The new version of the article that is provided by Sun et al. is still about tin oxide electron transport layer that is prepared by reactive sputtering. The authors mentioned in their response that the presented work is “very different" than the work reported by Qiu et al. [Scalable Fabrication of Stable High Efficiency Perovskite Solar Cells and Modules Utilizing Room Temperature Sputtered SnO2 Electron Transport Layer, Adv. Funct. Mater. 2018, 1806779]. The authors mentioned that their work is very different because they used a tin target instead of tin oxide as well as they used RF power source. However, I do not see any discussion or comparison between RF sputtering compared to DC sputtering or any advantages. There is only one sentence in the introduction, page 2, line 63 “radio frequency magnetron sputtering (RS) is a mature technology for low cost”. Similarly, no discussion at all about employing a tin target compared to tin oxide. Therefore, as I reported earlier, the presented research in this publication and Qiu publication is the same, no novelty, no addition, no enhancement or better results. The method of reactive sputtering is still the same. Also, same characterization methods were used.

In adittion, the authors compare their sputtered SnO2 with bare FTO to show the better PCE, EQE, PL, TRPL. It should have been compared with standard ETL layer, such as TiO2. The addition of any proper ETL would enhance the performance of the solar cell.

In addition, the article is still not properly written in many different parts (language and science), for example:

1.     Page 3, line 111, “AFM figures were measured using 300HVscanning force microscope (SEIKO)” . AFM here is first time used no prober abbreviation, and it is followed by “scanning force microscope” which would be SFM,…

2.     Same paragraph line 117 “ultraviolet photoelectron spectrum (UPS)” it should be spectroscopy.

3.     Line 120, “. a constant analyzer pass energy Ep Nine energy narrow scan 20eV, wide sweep 100eV”. I am not sure what the authors are trying to say here…

4.     In the results and discussions, page 3, line 131, the authors unnecessarily, continue describing their methods, “measured with a monochromatic He I light source (21.2 eV) and a VG Scienta R4000 analyzer. A sample bias of ‚-5 V was applied to observe the …”

5.     I would also highly recommend the authors to use values to describe their findings, such as, “valence band of the SnO2 prepared by reactive sputtering is much lower than the valence band of the perovskite”, how much?

6.     Lines 142-144, “compared with other literatures, it can be seen that the Sn peak shifts, indicating the presence of oxygen defects in the reactive sputtered SnO2”, how much does it shift and to what direction? I assume to higher binding energy in case we are comparing elemental Sn with oxidized Sn, but what are the values and how much is the difference?

7.     Line 144, “The vibration peak without Sn element appears, indicating that SnO2 can be prepared by reactive sputtering.” What vibration peak? Are the authors still talking about XPS of O1s core level?

8.     Figure 2 b, c, d, please use scientific notations for the Y axis, and either ways intensity of for UPS is an arbitrary unit, same applies for XPS.

9.     I would recommend the authors to zoom in for x axis, no need to show empty area, i.e. in fig 2 b no need to show below 2eV, or above ~6, in c no need for above 18 eV or below 15.5eV. Same applies for figure 3.

10.  Line 160, “The measured value of the roughness is the average value of the three non-passing areas on the surface of the sample” what are the three non passing areas?!

11.  Figure 7, no discussion or explanation about the figure, only in line 244 “This is consistent with our results in Figure 7.”

12.  The dark JV curve in figure 7 b is strange looking, it does not seem to correspond to JV curve of similar device under sun simulator (fig 7a).

Author Response

Dear Editor:

Many thanks to the reviewers for the insightful comments and suggestions concerning our manuscript, titled “Reactive sputtering tin oxide thin film as electron transport layer for planar perovskite solar cells” We have studied the comments carefully and have made corrections. Hopefully, these changes will make the manuscript more acceptable for publication.

Responds to the reviewer’s comments:

Review #3:

(1) The new version of the article that is provided by Sun et al. is still about tin oxide electron transport layer that is prepared by reactive sputtering. The authors mentioned in their response that the presented work is “very different" than the work reported by Qiu et al. [Scalable Fabrication of Stable High Efficiency Perovskite Solar Cells and Modules Utilizing Room Temperature Sputtered SnO2 Electron Transport Layer, Adv. Funct. Mater. 2018, 1806779]. The authors mentioned that their work is very different because they used a tin target instead of tin oxide as well as they used RF power source. However, I do not see any discussion or comparison between RF sputtering compared to DC sputtering or any advantages. There is only one sentence in the introduction, page 2, line 63 “radio frequency magnetron sputtering (RS) is a mature technology for low cost”. Similarly, no discussion at all about employing a tin target compared to tin oxide. Therefore, as I reported earlier, the presented research in this publication and Qiu publication is the same, no novelty, no addition, no enhancement or better results. The method of reactive sputtering is still the same. Also, same characterization methods were used.

Response: Thanks for the comment.

First of all, we did not copy the work of iu et al. Our work was completed in August and August, and the first draft was completed in June. We can show our raw data records. Below is a screenshot of our raw data.

The work of Qiu et al. was accepted in 2018. October. This is also the reason why we did not quote in the literature review at the beginning. When we finished the manuscript, the work of Qiu et al. was not published. We and Qiu et al. are paying attention to this research direction.

Second, questions about the innovative nature of our work. I hope the following articles on sputtering oxidation state will help you understand our work [1-4].

 [1] Rajmohan GD, Huang FZ, D'Agostino R, du Plessis J, Dai XJ. Low temperature reactively sputtered crystalline TiO2 thin film as effective blocking layer for perovskite solar cells. THIN SOLID FILMS 2017;636:307.

[2] Ge S, Xu H, Wang W, Cao R, Wu Y, Xu W, Zhu J, Xue F, Hong F, Xu R, Xu F, Wang L, Huang J. The improvement of open circuit voltage by the sputtered TiO2 layer for efficient perovskite solar cell. VACUUM 2016;128:91.

[3] Mali SS, Hong CK, Inamdar AI, Im H, Shim SE. Efficient planar n-i-p type heterojunction flexible perovskite solar cells with sputtered TiO2 electron transporting layers. NANOSCALE 2017;9:3095.

[4] Ke W, Fang G, Wang J, Qin P, Tao H, Lei H, Liu Q, Dai X, Zhao X. Perovskite Solar Cell with an Efficient TiO2 Compact Film. ACS APPL MATER INTER 2014;6:15959.

These four documents relate to RF and DC sputtering as well as titanium targets and titanium oxide targets.

Third, Why compare RF sputtering with DC sputtering because DC sputtering deposition rate is fast and target poisoning is easy. The RF sputter deposited film is relatively denser and flatter, but we should not easily conclude before we study the two sputtering methods in the next step. We respect the experimental results more.

In addition, since the tin target is a pure metal target, the target is easier to prepare due to the ductility of the metal during preparation, compared to the tin oxide target. Moreover, the tin target requires less excitation energy at the time of sputtering than the tin oxide target because the molecular weight of tin is smaller than the molecular weight of tin oxide. The combination of titanium oxide and nickel oxide sputtering in the application of perovskite batteries, so our work and Qiu et al. work is different.

We offer an additional preparation direction and are feasible. 

Finally, the detection methods are the same, because the current perovskite solar cells are mainly based on these detection methods. These detection methods are recognized as routine detection means. Therefore, most of the tests are the same, and this does not deny our work

(2) In adittion, the authors compare their sputtered SnO2 with bare FTO to show the better PCE,      EQE, PL, TRPL. It should have been compared with standard ETL layer, such as TiO2. The addition of any proper ETL would enhance the performance of the solar cell.

Response: Thanks for the comment. We are not comparing with standard titanium oxide, one is a different electron transport layer, and the other is because this is a different method.If we want to compare, it is better to compare the sputtered titanium oxide with the sputtered tin oxide. The work of sputtering titanium oxide is already in reference 1, with a PCE of 8.7%.From this point of view, the effect of our sputtered tin oxide is better than that of titanium oxide.

(3) Page 3, line 111, “AFM figures were measured using 300HVscanning force microscope (SEIKO)” . AFM here is first time used no prober abbreviation, and it is followed by “scanning force microscope” which would be SFM,…

Response: Thanks for the comment. Atomic Force Microscope    AFM, 

(4) Same paragraph line 117 “ultraviolet photoelectron spectrum (UPS)” it should be spectroscopy.

Response: Thanks for the comment. spectrum     spectroscopy

(5) Line 120, “. a constant analyzer pass energy Ep Nine energy narrow scan 20eV, wide sweep 100eV”. I am not sure what the authors are trying to say here…

Response: Thanks for the comment. This is the detection parameter of XPS.

(6) In the results and discussions, page 3, line 131, the authors unnecessarily, continue describing their methods, “measured with a monochromatic He I light source (21.2 eV) and a VG Scienta R4000 analyzer. A sample bias of ‚-5 V was applied to observe the …”

Response: Thanks for the comment. We have deleted this paragraph and amend it in the manuscript.

(7) I would also highly recommend the authors to use values to describe their findings, such as, “valence band of the SnO2 prepared by reactive sputtering is much lower than the valence band of the perovskite”, how much?

Response: Thanks for the comment. The band gap of tin oxide prepared by reactive sputtering is -4.04 eV to -7.64 eV, and the band gap of perovskite is -3.9 eV to -5.4 eV. Valence band of the SnO2 prepared by reactive sputtering is much lower than the valence band of the perovskite.

(8) Lines 142-144, “compared with other literatures, it can be seen that the Sn peak shifts, indicating the presence of oxygen defects in the reactive sputtered SnO2”, how much does it shift and to what direction? I assume to higher binding energy in case we are comparing elemental Sn with oxidized Sn, but what are the values and how much is the difference?

Response: Thanks for the comment. The peak position of Sn is shifted to the right by about 1 degree. There is a tin oxide prepared by standard solution method, which indicates that the sputtered tin oxide still has improvement. 14.6% is not the limit of reactive sputtering tin oxide.

(9) Line 144, “The vibration peak without Sn element appears, indicating that SnO2 can be prepared by reactive sputtering.” What vibration peak? Are the authors still talking about XPS of O1s core level?

Response: Thanks for the comment. We just want to show that there is no elemental tin present.

(10) Figure 2 b, c, d, please use scientific notations for the Y axis, and either ways intensity of for UPS is an arbitrary unit, same applies for XPS.

Response: Thanks for the comment. We have modified in the manuscript. It is showing it in the next answer.

(11) I would recommend the authors to zoom in for x axis, no need to show empty area, i.e. in fig 2 b no need to show below 2eV, or above ~6, in c no need for above 18 eV or below 15.5eV. Same applies for figure 3.

Response: Thanks for the comment.

We have modified in the manuscript.

(12) Line 160, “The measured value of the roughness is the average value of the three non-passing areas on the surface of the sample” what are the three non passing areas?

Response: Thanks for the comment. It means three different positions of a sample.

(13) Figure 7, no discussion or explanation about the figure, only in line 244 “This is     consistent with our results in Figure 7.”

Response: Thanks for the comment. It can be seen from Fig. 7c that the carrier transport of the perovskite film containing the tin oxide layer is faster than that of the perovskite layer without the tin oxide layer, further demonstrating that the tin oxide prepared by reactive sputtering is suitable as the electron transport layer. This is consistent with the results of PL. In addition, the CV curve of the dark state CV curve and the illumination condition are also consistent.

(14) The dark JV curve in figure 7 b is strange looking, it does not seem to correspond to JV curve of similar device under sun simulator (fig 7a).

Response: Thanks for the comment. It is the detection of the same sample. We first examine the CV curve, then the dark CV curve, which may have some effect.

We tried our best to improve the manuscript and made some changes in the manuscript. These changes will not influence the content and framework of the paper. We did not list the changes but marked in red font in revised paper.

Once again, thank you very much for your comments and suggestions.

Round  3

Reviewer 1 Report

The revised manuscript has answered all the comments. 

Author Response

Thank you for your review and we have provided valuable comments for us, which is very important for us to improve the quality of the manuscript. Thank you again for your review.

Reviewer 3 Report

I would like to thank the authors for their response and highlight that my comments and questions to them as a referee are not personal but to improve the quality of manuscript for community. Similar questions will be expected from the readers. I would recommend the authors to use the provided time by the editor for revision efficiently, to improve the article as much as possible. Therefore avoid many rounds or revision. This can be implemented at every round by reading the whole manuscript thoroughly and carefully and by considering all the comments from the referees to be incorporated in the new version, rather than just in the response letter. Therefore I would like to highly recommend the authors to incorporate these answers in the new version of the manuscript. Please find below more details.

·      In the response letter, page 2-3, I would highly recommend the authors to incorporate the response in the article with supporting references including Qiu et al. article:

“…Why compare RF sputtering with DC sputtering because DC sputtering deposition rate is fast and target poisoning is easy. The RF sputter deposited film is relatively denser and flatter, but we should not easily conclude before we study the two sputtering methods in the next step. We respect the experimental results more.

In addition, since the tin target is a pure metal target, the target is easier to prepare due to the ductility of the metal during preparation, compared to the tin oxide target. Moreover, the tin target requires less excitation energy at the time of sputtering than the tin oxide target because the molecular weight of tin is smaller than the molecular weight of tin oxide. The combination of titanium oxide and nickel oxide sputtering in the application of perovskite batteries, so our work and Qiu et al. work is different.

We offer an additional preparation direction and are feasible. Finally, the detection methods are the same, because the current perovskite solar cells are mainly based on these detection methods. These detection methods are recognized as routine detection means. Therefore, most of the tests are the same, and this does not deny our work

·      Page 4, pint (5), please correct:

o   pass energy Ep –> pass energy (Ep)…

o   for the narrow scans (high resolution), Is Ep 9 eV or 20??. I think that the authors mean that Ep for the high resolution scan was 20eV, and for survey (wide) it was 100 eV, but what is “Nine”??! also there is no survey scan provided, so please remove 100ev for wide scan.

·      Point (7), please write in the manuscript that VB of SnO2 ….. is lower by X.XX value instead of only saying “much lower”. Same applies in general for the manuscript, please always consider using values.

·      Point (8):

o   …about 1 degree… !!!??, this is wrong, it should be 1 eV, this is XPS not XRD. In XPS there is energy of the electrons not diffraction angle in degrees. Please correct!

o   Also please provide the value of the binding energy (BE) from the cited reference for the peak maximum as well as the BE peak maximum from your measurement.

o   If the BE measured by the authors is shifted to right (I guess the authors mean to lower binding energies), then that is elemental Sn, or Sn with lower oxidation state not Sn oxide. Once the authors provide the value from literature that they mentioned then it is easier to say or see that there is a shift to lower BEs or to higher BEs.

o   Please incorporate and revise the text in your manuscript accordingly.

·      Point (9):

o   XPS is NOT a spectroscopy that is based on vibrations! you cannot say "the vibration peak". In XPS kinetic energies of different excited electrons from the core levels are measured.

o   Please revise the text in the manuscript accordingly.

·      Point (12), please revise the text to make it clear that you mean three different positions.

·      Point (14), please incorporate in the manuscript and elaborate.

Author Response

Responds to the reviewer’s comments:

Review #3:

(1)  In the response letter, page 2-3, I would highly recommend the authors to incorporate the response in the article with supporting references including Qiu et al. article:

Response: Thanks for the comment. We have already added this document to the introduction.

(2) Page 4, pint (5), please correct:  pass energy Ep –> pass energy (Ep)…

    for the narrow scans (high resolution), Is Ep 9 eV or 20??. I think that the authors mean that Ep for the high resolution scan was 20eV, and for survey (wide) it was 100 eV, but what is “Nine”??! also there is no survey scan provided, so please remove 100ev for wide scan.

Response: Thanks for the comment. a constant analyzer pass energy (Ep) energy narrow scan 20eV, We have modified and removed 100eV

(3) Point (7), please write in the manuscript that VB of SnO2 ….. is lower by X.XX value instead of only saying “much lower”. Same applies in general for the manuscript, please always consider using values.

Response: Thanks for the comment. The valence band of the SnO2 prepared by reactive sputtering is lower by-5.4eV. This shows that the sputtered tin oxide is suitable for the electron transport layer

(4)   …about 1 degree… !!!??, this is wrong, it should be 1 eV, this is XPS not XRD. In XPS there is energy of the electrons not diffraction angle in degrees. Please correct!

   Also please provide the value of the binding energy (BE) from the cited reference for the peak maximum as well as the BE peak maximum from your measurement.

     If the BE measured by the authors is shifted to right (I guess the authors mean to lower binding energies), then that is elemental Sn, or Sn with lower oxidation state not Sn oxide. Once the authors provide the value from literature that they mentioned then it is easier to say or see that there is a shift to lower BEs or to higher BEs.

 Please incorporate and revise the text in your manuscript accordingly.

 Response: Thanks for the comment. the binding energies of the Sn 3d5 / 2 and Sn 3d3 / 2 peaks in the literature are 487.11 and 495.56 eV. However, the binding energies of tin oxide prepared by reactive sputtering are 495.14 and 486.69 eV, respectively. The tin oxide binding energy prepared by reactive sputtering is 0.42 eV lower than that in the literature. It shows that the tin oxide prepared by reactive sputtering contains a part of stannous, which is not completely tin oxide.

(5)  XPS is NOT a spectroscopy that is based on vibrations! you cannot say "the vibration peak". In XPS kinetic energies of different excited electrons from the core levels are measured.

   lease revise the text in the manuscript accordingly.

Response: Thanks for the comment. We have removed it. Corrected in the manuscript

(6)  Point (12), please revise the text to make it clear that you mean three different positions.

Response: Thanks for the comment. We have modified in the manuscript.

(7)  Point (14), please incorporate in the manuscript and elaborate.

Response: Thanks for the comment. We have modified in the manuscript.

    Thank you very much for your review. What you pointed out is something we didn't pay attention to before, which is very important for improving our manuscript. Thank you again for your review.

We tried our best to improve the manuscript and made some changes in the manuscript. These changes will not influence the content and framework of the paper. We did not list the changes but marked in red font in revised paper.

Once again, thank you very much for your comments and suggestions.

Coatings EISSN 2079-6412 Published by MDPI AG, Basel, Switzerland RSS E-Mail Table of Contents Alert
Back to Top